# Optimizing Input Feature Sets Using Catch-22 and Personalization for an Accurate and Reliable Estimation of Continuous, Cuffless Blood Pressure

**DOI:** 10.3390/bioengineering12050493

**Published:** 2025-05-06

**Authors:** Rajesh S. Kasbekar, Srinivasan Radhakrishnan, Songbai Ji, Anita Goel, Edward A. Clancy

**Affiliations:** 1Department of Biomedical Engineering, Worcester Polytechnic Institute (WPI), Worcester, MA 01609, USA; sji@wpi.edu (S.J.); ted@wpi.edu (E.A.C.); 2Department of Mechanical and Industrial Engineering, Northeastern University, Boston, MA 02115, USA; srk.srinivasan@gmail.com; 3Nanobiosym Inc., Cambridge, MA 02142, USA; agoel@nanobiosym.com; 4Department of Physics, Harvard University, Cambridge, MA 02138, USA; 5Department of Electrical and Computer Engineering, Worcester Polytechnic Institute (WPI), Worcester, MA 01609, USA

**Keywords:** cuffless blood pressure estimation, nocturnal monitoring, machine learning, deep learning, transfer learning, photoplethysmogram waveform

## Abstract

Nocturnal monitoring of continuous, cuffless blood pressure (BP) can unleash a whole new world for the prognostication of cardiovascular and other diseases due to its strong predictive capability. Nevertheless, the lack of an accurate and reliable method, primarily due to confounding variables, has prevented its widespread clinical adoption. Herein, we demonstrate how optimized machine learning using the Catch-22 features, when applied to the photoplethysmogram waveform and personalized with direct BP data through transfer learning, can accurately estimate systolic and diastolic BP. After training with a hemodynamically compromised VitalDB “calibration-free” dataset (n = 1293), the systolic and diastolic BP tested on a distinct VitalDB dataset that met AAMI criteria (n = 116) had acceptable error biases of −1.85 mm Hg and 0.11 mm Hg, respectively [within the 5 mm Hg IEC/ANSI/AAMI 80601-2-30, 2018 standard], but standard deviation (SD) errors of 19.55 mm Hg and 11.55 mm Hg, respectively [exceeding the stipulated 8 mm Hg limit]. However, personalization using an initial calibration data segment and subsequent use of transfer learning to fine-tune the pretrained model produced acceptable mean (−1.31 mm Hg and 0.10 mm Hg) and SD (7.91 mm Hg and 4.59 mm Hg) errors for systolic and diastolic BP, respectively. Levene’s test for variance found that the personalization method significantly outperformed (*p* < 0.05) the calibration-free method, but there was no difference between three machine learning methods. Optimized multimodal Catch-22 features, coupled with personalization, demonstrate great promise in the clinical adoption of continuous, cuffless blood pressure estimation in applications such as nocturnal BP monitoring.

## 1. Introduction

Cuffless blood pressure (BP) estimation based on continuous measurements, e.g., using the photoplethysmography (PPG) and electrocardiogram (ECG) signals rather than the intermittent cuff-based measurement of BP, has gathered significant attention recently [1]. PPG is an optical measurement of tissue underneath the sensor and reflects heart activity non-intrusively and non-invasively [2]. While the intent of such BP monitoring is to minimize patient discomfort and harness the significant advantages of continuous measurement compared with intermittent clinical cuff BP measurements taken during the day, there is a more compelling reason to remove any barriers related to its clinical adoption: nighttime or nocturnal BP.

Continuous measurement of nocturnal BP is a far better predictor of cardiovascular disease and stroke than daytime BP [3]. Furthermore, nocturnal hypertension can occur in people whose daytime BP is normal [3]. Spikes in BP during sleep can have potentially serious health implications that include heart failure and other such cardiovascular or renal diseases. Nighttime BP, therefore, has strong prognostic value in predicting outcomes. It allows for the diagnosis of masked hypertension due to isolated nocturnal hypertension. It offers information on cardiovascular modulation during sleep for both healthy and diseased conditions. In addition, monitoring BP during the night can help measure the effect of drug- or device-based treatment over a 24 h period [4]. Nonetheless, nocturnal hypertension is not easy to measure as routine BP checks are mostly taken during daytime hours. Because it is not practical to make frequent cuff-based BP measurements at night, an accurate and reliable method for cuffless and continuous BP measurement is critical for the measurement of nocturnal BP.

Measurement of cuffless, continuous BP using PPG has, however, been fraught with a myriad of challenges [5,6]. Clinical adoption of cuffless devices has been slow due to the issue of trusting the measurement in the individual patient [7,8]. To address this need, the European Society of Hypertension created revised recommendations [9] for the validation of cuffless BP measuring devices, with the goal of providing a standard for clinical acceptance of such devices. These recommendations include six stepwise device-specific tests depending upon device type and include both static and dynamic tests as well as tests based on device position, exercise, treatment, sleep, and re-calibration. They are based on a comprehensive, demanding, and complex methodology built on established principles from the American Association of Medical Instrumentation (AAMI), The International Organization for Standardization (ISO), and European Society of Hypertension.

Various machine learning (ML) and deep learning (DL) techniques have been utilized on measures associated with systolic blood pressure (SBP), diastolic blood pressure (DBP), and mean arterial pressure (MAP), including PPG, bioimpedance, tono-arteriogram (TAG) or other waveforms, pulse rate, and demographic data. Zhao et al. [1] comprehensively reviewed the enabling technologies for wearable and cuffless BP platforms for continuous as well as “beat to beat” BP measurement that include sensor designs, pulse transit time-based analytical models, and ML algorithms. They compared the various multimodal input modalities, features, implementation algorithms, and performances to monitor BP. They concluded that measurement reliability can only be achieved by evaluating sufficient population size with intra- and inter-individual BP diversity. They suggested that continuous BP monitoring (with TAG tracking) should be a key area of focus in the field. The various ML and DL algorithms recently used to estimate SBP, DBP, and MAP using cuffless, continuous approaches are shown in Table 1. While some studies found bias and standard deviation (SD) errors that met the AAMI criteria (of <5 mm Hg bias and <8 mm Hg SD, respectively), these studies had some major data biases. They were either prospective studies conducted on healthy subjects or they had selective pre-processing that eliminated outliers and poor quality signals that are inherent in any PPG or ECG signal. Furthermore, all of these studies were short term, ranging from 10 s to a maximum of a few hours (as shown in Table 1), such that it was not possible to verify the method over a longer time period, which is critical for continuous monitoring. None of the studies therefore accounted for longer-term variations in BP that could result in long-term drift errors.

Accurate and reliable estimation of BP in a cuffless device that is reliable for long time periods is challenging, with SBP proving more difficult to estimate than DBP (see Table 1). Previously [27], we laid out the theory behind cuffless BP estimation using demographic information and features extracted from (i) pulse arrival time (PAT) or pulse transit time (PTT) and (ii) pulse wave morphology (PWM). We demonstrated that the use of a high-quality PPG signal [28,29], selective features, and an optimal machine learning algorithm can provide promising results in estimating SBP and DBP. The motivation for the present paper, therefore, was to extend to a reliable long-term, clinically acceptable method using a three-fold strategy: (i) to evaluate performance improvements when using over 1500 subjects from the VitalDB (PulseDB) database; (ii) to investigate a state-of-the-art feature extraction algorithm called Catch-22 (CAnonical Time-series Characteristics), selected through highly comparative time-series analysis on PPG waveforms; and (iii) to measure the impact of targeted personalization by using the SBP and DBP obtained from the test subject as personalization and feed it back into the general training algorithm through transfer learning. Catch-22 is a method of extracting a relatively small set of relevant features from the pantheon of features utilized in time-series analysis. This small set of features (including linear and non-linear autocorrelation, successive differences, value distributions and outliers, and fluctuation scaling properties) has exhibited strong classification or regression performance in past studies and is minimally redundant. Personalization is a calibration that fine-tunes the general models using a small segment of the target subject’s data consisting of the mean and SD of the BP signal. The data are used for calibration in the training dataset and for insertion into the the main algorithm layers using transfer learning. Alternatively, a calibration-free method [22] also exists that does not use any target subject data for training or insertion into the main algorithmic layers. Our hypothesis is that the use of this large dataset, along with the selective features on key waveform inputs and personalization, will significantly enhance the accuracy for SBP and DBP, comparing with intra-arterial BP as the gold standard and ensuring its reliability for continuous application. We tested for statistically significant differences in error means as well as variances between the two calibration methods (personalization versus calibration-free) and three training algorithms (Lasso, random forest and ResNET).

The European Society of Hypertension working group on blood pressure monitoring and cardiovascular variability issued a statement in 2022 [7] recommending at that time against the adoption of devices estimating BP using cuffless methods in clinical practice. According to their consensus statement, although cuffless sphygmomanometers are a promising technology, standardization of accuracy, tracking of dynamic variations, and suitability for long-term use is required—and the existing literature had not demonstrated these characteristics. For widespread use in clinical practice, evidence is needed not only for measurement in healthy subjects but also in pre-hypertensive and hypertensive patients to ensure that the system can be used in the same way as, or instead of, current blood pressure monitors. Therefore, until a device is developed that can meet these requirements, its use in clinical practice cannot be recommended. We adopted a different approach to address this exact issue. Our key contribution, therefore, is based on the premise that the SBP and DBP, estimated non-invasively by our method, is reliable only if the flow or the pulse arrival time is within a certain quantum band. A change in flow or pulse arrival time outside this band triggers a personalization or calibration that would ensure that accuracy is preserved. Any creep due to confounding factors contributing to a large BP variance, especially over longer time periods, would no longer be relevant in increasing such inherent variance. This approach ensures that the accuracy and reliability of the BP estimates is maintained over the long term for hypertensive or hypotensive subjects and therefore could be suitable for clinical adoption.

This paper is organized in the following manner. The Methods section describes the experimental methods consisting of the patient data, feature extraction, method of analysis consisting of the machine and deep learning model training with hyperparameter selection, and method of analysis consisting of the personalized and calibration-free model testing and statistical analysis. The Results section shows the descriptive and statistical data results comparing the two calibration methods and three algorithms. Finally, the Discussion section elaborates on the key insights obtained from the data, the limitations, and future research directions.

## 2. Methods

### 2.1. Human Data Ethical Statement

This study utilized data from the publicly available VitalDB dataset [30]. All subjects are deidentified. Human studies permissions and procedures, including written informed consent, were the responsibility of the data originators. Data acquisition and disclosure were approved by the Institutional Review Board of Seoul National University Hospital (H-1408-101-605), and the study was registered at clinicaltrials.gov (NCT02914444).

### 2.2. Patient Data

Wang et al. [30] pre-processed and extracted data from the larger PulseDB database to create the VitalDB database. The PulseDB database is the largest filtered and cleaned database to date that enables a standardized, reliable, and reproducible evaluation of cuffless BP estimation models [31]. VitalDB contains hemodynamically compromised patient recordings (with demographic information) of continuous waveform ECG, PPG, and ABP data, sampled at 125 Hz, recorded over 5 to 10 days from ICU patients who underwent surgeries in the Seoul National University Hospital, South Korea. The criteria used by Wang et al. [30] for extraction included the presence of ECG lead-II, fingertip PPG, and ABP signals. Wang et al. [30] extracted, filtered, and cleaned data for invalid samples, saturated and flatline signals, and quality of the signals (see [30], Section 2.4 and Section 2.5). The R-wave peaks of the ECG signals were detected for each record using the Pan–Tompkins QRS detection algorithm [32]. Systolic peaks of the PPG signal were located using Elgendi’s algorithm [33], and diastolic valleys were located as the minimum between every two consecutive systolic peaks. Next, Wang et al. [30] divided each signal (ECG, PPG, and ABP) into contiguous 10 s segments that were, on average, 9 min apart. Segments having more than three consecutive samples of the same value equaling the minimum or maximum amplitude within the segment, or more than 1 s of the same amplitude (i.e., saturated/flatlined) for any signal, were removed. Reference SBP and DBP values of each segment were defined as the average beat-to-beat SBP and DBP values within that segment. This pre-processing, extraction and signal processing was implemented by [30]. We further removed the ECG, PPG, and ABP signals with less than 3 peaks.

Wang et al. [30] then assembled a subset of 1525 subjects that met the American Association of Medical Instrumentation (AAMI) test data inclusion criteria to ensure a uniform representation of hypotensive and hypertensive subjects. The 1525 subjects were sub-divided by Wang et al. [30] into 1293 subjects for training, 116 for validation, and 116 for testing. The AAMI criteria [34] were applied to the test dataset and mandate at least 85 subjects in this test dataset. At least 5% of these subjects must belong to each of the following categories: an SBP below 100 mm Hg, an SBP above 160 mm Hg, a DBP below 60 mm Hg, and a DBP above 100 mm Hg. In addition, at least 20% of the subjects must each have an SBP over 140 mm Hg and a DBP over 85 mm Hg. The test dataset met these criteria. Per subject, we randomly selected 19, 7, and 7 “10-s segments” for each of training, validation, and testing, respectively. The 19 segments were randomly chosen because for 1293 subjects, they gave a total of 24,567 data segments, which were well above fit parameter requirements (10 times 840 fit parameters or 8400) of the deep learning ResNET algorithm. As detailed below, only 3 test values per subject were evaluated for estimation error from the 7 available test segments as stipulated by the AAMI criteria. Seven segments were randomly chosen and arranged in numeric order because the first segment was used for personalization, 3 test values per subject were required by the AAMI criteria, and 3 additional data segments had to be extracted to maintain independence of the direct pressure calibration values if the flow criteria were not met in the initial 3 segments, as explained in Section 2.5 below.

### 2.3. Feature Extraction

From every 10 s data segment, we extracted the “Catch-22” set of 22 features (see Appendix A) [35]. This set includes a generation of small, canonical subsets of features that display high performance across a given ensemble of tasks. They also exhibit complementary performance characteristics with each other. Catch-22 uses linear and non-linear autocorrelation, successive differences, value distributions, outliers, and fluctuation scaling to extract features that reduce dimensionality, are minimally redundant, and facilitate feature-based time-series analysis.

In addition to the Catch-22 features, we also extracted the following features: the mean and SD of the PPG, two temporal features (PAT and heart rate), and four PPG morphology features. The PAT was extracted from each data record by computing the average of the first 3-time intervals between the ECG R-peak to the ensuing PPG valley. The heart rate was derived from the average time interval for the same three consecutive ECG R-peak to R-peak intervals. The four PPG morphology features were extracted as an average from three successive beats, randomly selected. The features were the average time interval over 3 beats between the valley and peak of the PPG waveform, the average time interval over 3 beats between the peak of the same PPG waveform and valley of the subsequent PPG waveform, the peak amplitude over 3 beats, and the derivative of the PPG peak amplitude computed as the average of the slope of the line connecting the valley to the peak over 3 beats. We also used the demographic features of age and sex, giving a total of 32 input features for the calibration-free method. For the personalized dataset only, we also utilized in a unique manner (see below) the two additional features of ABP mean and standard deviation, as the VitalDB data did not contain any cuff BP measurements. In practice, a systolic and diastolic reading obtained from a cuff, or any other reliable direct BP measurement method, would be used to substitute for these ABP features. This resulted in the use of a total of 34 input features for personalization. These same set of features were analyzed for relevance using Shapely analysis and then used in all three ML algorithms to enable a comparison of the performance among the three algorithms. Figure 1 shows a schematic diagram of the flow steps consisting of feature extraction, training, hyperparameter selection, and testing.

### 2.4. Method of Analysis—Machine Learning Model Training

We trained three model forms on the training data (unless noted otherwise below), Lasso (linear regression with ML), random forest (ML) and ResNET (DL), for each of SBP and DBP estimation using the ML and DL toolbox from MATLAB R2023. These model forms include varying levels of complexity and number of fit parameters (Lasso having the lowest and ResNET the highest). They are widely used for predictions ranging from Lasso, which has the lowest number of fit parameters (100), to random forest (492) and ResNET with the highest number of fit parameters (840). Lasso is an extension of the linear regression model, while random forest and ResNET use alternate approaches based on non-linear decision-making that work well in predictive applications. Table 2 below shows a comparison between the three methods.

Lasso (least absolute shrinkage and selection operator) is a regression analysis method that chooses key features and performs regularization to prevent overfitting and enhance the prediction accuracy and interpretation of the resulting model. Lasso extends the ordinary least squares regression by adding a penalty term to the regression equation. The penalty term is the sum of the absolute values of the coefficients, which helps in shrinking some coefficients to zero, effectively performing feature selection. Lasso uses an elastic net hyperparameter (“alpha”), an estimate of the Lasso to ridge variance, which controls the strength of the penalty.

For our Lasso models (Lasso command in MATLAB), BP prediction was based on a linear model using least squares regression and 10-fold cross-validation. The elastic net hyperparameter (“alpha”) was selected as 0.75 based on our prior experience [27]. The loss curves from the validation dataset as well as the bias and variance of the validation dataset were used post hoc to confirm these hyperparameters.

Random forest is an ensemble learning method (which we used for regression) that works by bagging and averaging a multitude of decision trees using a random subset of features from the main feature set. The training algorithm applies the bagging technique to tree learners as follows: Given a training set with multiple responses, bagging repeatedly “n” times selects a random sample with replacement of the training set and fitting trees to these samples. After these “n” trees are trained, the predictions are averaged from all of the individual regression trees.

For random forest models (random forest, fit ensemble command in MATLAB), prediction was based on a trained regression ensemble model using 10-fold cross-validation that included boosting several regression trees. We used the built-in MATLAB hyperparameter optimization function (OptimizeHyperparameters’, ‘all’) that cycled through the lag and boost variations as well as the learning cycles, learning rate, minimum leaf size and maximum number of splits, and number of sample variables to identify the most optimal combination on the training dataset. This optimization led to the following hyperparameters: LSBoost method, a learning rate of 0.22, 463 learning cycles, leaf size of 2 with 12 splits, and 5 variables to sample. The loss curves from the training and validation dataset as well as the bias and variance of the independent validation dataset were used to confirm these hyperparameters.

For ResNET (using the deep learning application in MATLAB), the prediction used the architecture shown in Figure 2. ResNET is a convolutional neural network architecture that is 18 layers deep. It consists of an input layer and a fully connected layer; four processing stages, each composed of a batch normalization layer, a ReLu layer, and a fully connected layer; a fifth processing stage consisting of batch normalization layer and a fully connected layer; two addition convolution neural network layers; and an output regression layer. The model skips layers during training, allowing it to train easily and achieve better accuracy. The number of epochs was chosen to be 50 as the performance saturated after 50 epochs, with minimal improvement in performance, when tested on the validation dataset from 20 to 500 epochs. We used 5-fold cross-validation on the training data and the final chosen hyperparameters included 50 epochs and a learning rate of 0.001. The loss curves from the training and validation dataset as well as the bias and variance of the validation dataset were used post hoc to confirm these hyperparameters.

### 2.5. Method of Analysis—Personalized and Calibration-Free Model Testing

The 7 randomly selected test segments in the test dataset were placed in sequential order, 1, 2, 3, …, 7. Not all segments reserved for testing were used as such, and segment 1 from each subject was designated as the personalization segment that was never used for testing. Personalization consisted of the ABP mean and SD features, whose values were always taken from a data segment occurring sooner in time than the test segment, from the same subject. By default, the mean and SD of the ABP values for test segment 1 (the personalization segment) were set aside. Then, test segment 2 was paired with 5, 3 with 6, and 4 with 7. If the mean PPG value from segment 2 differed by ≤5% from the mean PPG value of segment 1, then the default mean and standard deviation ABP from segment 1 was the personalization input for segment 2, and segment 5 was unused for testing. Otherwise, the ABP mean and ABP standard deviation from segment 2 were the personalization inputs for segment 5 (making segment 2 the personalization trial), and segment 2 was unused for testing. (We refer to this switch in the personalization trial as re-calibration.) The other 32 input features always came from the test segment used. Similar conditional testing was performed on the two other paired segments, with segment 1 always being used as the default. Three test results were thus computed per subject. This personalization ensured that subject-specific information was inserted as an input layer to the algorithm. The ABP mean and SD values from a test segment were never used when testing that segment. For calibration-free models, testing using the 32 input features was conducted on the same 3 segments per subject, for which testing was performed using the personalized method. Doing so balanced the test dataset between the personalized and calibration-free methods.

Note that a cutoff of a 5% change in the mean PPG signal in personalization was selected empirically by evaluating the estimated systolic BP error SD using the ResNET algorithm as the mean PPG threshold was varied. The systolic error SD for change in mean PPG by 4%, 5%, and 6% was found to be 6.56 mm Hg, 7.91 mm Hg, and 8.34 mm Hg, respectively. Thus, a 5% change in mean PPG was used as it gave a systolic error SD <8 mm Hg as required by the AAMI criteria.

### 2.6. Statistical Analysis

For each combination of the ML algorithm and model training approach (personalized, calibration-free), we measured the test data bias (mean error or ME), mean absolute error (MAE), and SD error between estimated SBP and, separately, DBP in comparison with the corresponding labelled ABP data.(1)$$ME\;(Mean\;Error)=\frac{1}{n}.\sum_{i=1}^{n}\left(Âi-Ai\right)$$(2)$$SD\;(Standard\;Deviation)=\sqrt{\frac{1}{n-1}.\left[\sum_{i=1}^{n}{\left(\left(Âi-Ai\right)-ME\right)}^{2}\right]}$$(3)$$MAE\;(Mean\;Absolute\;Error)=\frac{1}{n}.\sum_{i=1}^{n}\left|(Âi-Ai)\right|$$(4)RE (Residual Error) = (Âi−Ai)
where n is the number of total estimations, Ai is the ith reference SBP or DBP value, and Âi is the ith SBP or DBP value estimated by the model.

ME and SDE are estimators of the BP estimation bias, and the range of errors in which the model’s error on the population resides, under the assumption of normally distributed residual errors. Because mean arterial pressure (MAP) is correlated with SBP and DBP, we did not show it separately. We also computed the number of residual errors below 5 mm Hg, 10 mm Hg, and 15 mm Hg as required by the British Society of Hypertension standard for BP monitoring. Two-way analysis of variance (ANOVA) with post hoc multiple comparison tests were used to test significant differences in the means (for each of the 3 ML algorithms), per subject per case (for each of the two calibration methods). Levene’s test was similarly used to test for significant differences in the variances. In all tests, *p* < 0.05 was considered as significant.

## 3. Results

Demographic information for the utilized VitalDB subjects is reported in Table 3. At least 40% of the subjects were female, and over 85% were above 40 years of age. Model re-calibration is highly dependent on the hemodynamically compromised nature of the subject data and was required for over 35% of the subjects based on our data sample. The average time duration between the personalization data segment and the first test data segment was 25.6 h; between the personalization and second test data segments was 26.2 h; and between the personalization and third data segments was 26.6 h.

Table 4a,b show summary estimation errors for SBP and DBP for the three algorithms and the two calibration methods. For both the calibration-free dataset as well as the personalization dataset, each bias error easily met criterion 1 of the IEC 80601-2-30 [34] standard of ≤5 mm Hg. As shown in Table 4a, none of the SDs for SBP or DBP using the calibration-free method met the AAMI requirements of the AAMI/IEC 80601-2-30 [34] standard. However, the SDs for both the SBP and DBP using the personalization method met these requirements when used with the random forest and ResNET methods. Bland–Altman plots [36] for the SBP and DBP using the ResNET algorithm (which had the lowest SD error) are shown in Figure 3.

Furthermore, Table 4c shows the performance of the ResNET model with the British Hypertension Society (BHS) standard [37], which assigns grades to performance based on the percentage of readings below 5 mm Hg, 10 mm Hg, and 15 mm Hg. The ResNET method received grade A for both SBP and DBP.

A two-way analysis of variance for statistical differences in bias errors (separately for each of SBP and DBP) was conducted using the test segments of the 116 subjects compared between the three ML algorithms and two calibration methods. For each of SBP (*p* = 0.97) and DBP (*p* = 0.99), ML algorithm–calibration method interactions existed. Thus, we proceeded to paired-multiple comparisons (separately for SBP and DBP) between each ML algorithm–calibration method combination. All comparisons were not significant for SBP (*p* ≥ 0.29) and for DBP (*p* ≥ 0.23).

Levene’s test is a statistical method to assess the difference between the variances of two independent datasets. It uses the mean absolute error to test the null hypothesis that the variances of two datasets estimated using different methods or using different algorithms (also referred to as the homogeneity of variance or homoscedasticity) are equal. A *p*-value of less than 0.05 indicates a difference between the variances in the datasets assuming normal distribution and random sampling. Our objective in using this test was (i) to infer statistically whether the variance of the data estimated using the personalization method differed from the variance of data estimated using the calibration-free method; and (ii) to infer statistically whether the variance of the data estimated using the three ML algorithms varied from each other.

Table 5 presents the result of Levene’s absolute test for the statistical difference in variances (based on absolute value and separately for SBP and DBP) comparing the three ML algorithms and two calibration methods using the first utilized test segment of the 116 subjects. For both SBP and DBP, the calibration method used had a significant impact on the variances, while the ML algorithm used showed no such impact. Post hoc paired multiple comparisons with Bonferroni correction conducted for the three algorithms for both SBP and DBP found that the personalization method was significantly better than the calibration-free method for variance (*p* ≤ 0.00).

We next wanted to investigate if subject BP would be better estimated by the most recent “ground truth” measurement, absent of any information provided by physiological measurements. Thus, for each subject, we computed the error between SBP (and, separately, DBP) from each ABP signal of each personalization segment vs. the ABP signal of their paired test segments. Again, ABP signals are used herein as surrogates for BP cuff measurements. Table 6 also shows statistical results comparing the “ground truth” errors with those found when using the ResNET algorithm. There was no difference in bias error for either SBP (*p* = 0.26) or DBP (*p* = 0.05). However, the values estimated by our algorithm exhibited significantly lower variance error for each of SBP (*p* = 0.00) and DBP (*p* = 0.00).

## 4. Discussion

In our previous paper [27], we presented a theoretical framework for estimation of continuous BP using cuffless devices. In this publication, our goal was to apply this theoretical framework on a large dataset to see the impact of scale as well as feature selection and personalization (using mean PPG flow change as the re-calibration criterion) in enhancing the BP estimates. We utilized the standardized and reliable VitalDB dataset, containing hemodynamically compromised (diseased) subjects having a wide range of BP values including abnormal BP values, to see the impact of scaling sample size, optimal feature extraction, and personalization using this framework.

We applied a unique Catch-22 feature extraction algorithm [35] on the PPG and ECG waveform data, and along with temporal (pulse arrival time and heart rate), demographic, and morphology data, we estimated the SBP and DBP using a calibration-free method. Despite optimizing the hyperparameters for each algorithm, our SD results with the Lasso, RF, and ResNET methods were outside the 8 mm Hg SD limits of the BP performance standard criterion 1 [34]. These results showed that ML or DL algorithms trained on even larger scale datasets using the features, signals, and models we used could not fully learn them to reduce the variability within acceptable limits for commercial deployment, due to the large variability created by the confounding factors.

We then explored the use of personalization to reduce SD errors within the model. Personalization involves fine-tuning the model using ”subject specific” data and has been used by others. Schrumpf et al. [38] used transfer learning to fine-tune a specific layer of their model with a small amount of template data from the target subject. Leitner et al. [39] used transfer learning that personalized specific network layers to reduce the number of required training samples, which further improved the performance of the BP estimation. Bresch et al. [40] used single-parameter personalization on the performance of multi-parameter models, which significantly enhanced continuous BP tracking. We extracted the mean and SD from a personalization ABP data segment to fine-tune the pretrained architecture of the main general model classifier and tested the model on distinct test set data segments. A 5% change in the mean PPG signal was then used as a criterion to re-calibrate or re-personalize the general model classifier. The use of this criterion ensured that re-calibration had to be performed only upon a presumed change in BP, as characterized by a change in the mean PPG signal, rather than on a periodic basis. The PAT could alternatively be used in place of the mean PPG signal as the criteria for re-calibration. The anticipated frequency of this re-calibration likely depends on the extent of disease associated with the hemodynamic compromise of the PPG flow or BP values. For example, healthy subjects with minimal disruption in PPG flow and BP would be expected to require re-calibration after a much longer time, while diseased subjects might need it more frequently. Because our test dataset had a population with significant disease conditions, over 35% needed a re-calibration approximately after every 24 h on average. This personalization technique using the random forest or ResNET algorithm significantly reduced error variance to within the AAMI limits (Table 4b) as well as resulting in a BHS grade A performance (Table 4c). It also ensured that this variance and performance would not be affected by any long term drift or change in the hemodynamic state of the subject.

One key limitation of our method was that we used the arterial BP waveform from the VitalDB dataset for personalization. A pressure-related waveform such as the TAG waveform should also be explored instead of cuff measurements as a substitute for this arterial BP waveform. In their review paper of emerging technologies for continuous and cuffless BP monitoring, Zhao et al. [1] concluded that TAG tracking will be the next wave of focused research in continuous BP monitoring. TAG sensors [41] are flexible sensors that can lodge on curved or soft surfaces without causing discomfort. Therefore, they can measure continuous, cuffless pulsations transmitted from the BP in the vascular wall of arteries to the skin surface. These sensors capture a pulse waveform comprising the (i) percussion-wave (or P wave), caused by the systolic pressure spike due to blood ejection from the contracting left ventricle [42]; (ii) tidal wave (or T-wave), caused by the BP reflection from the upper body; and the (iii) diastolic wave (or D-wave), caused by the BP reflection from the lower body [43]. The TAG waveform with appropriate calibration for scaling could be a convenient substitute for the arterial BP waveform in personalizing the subject-specific data for estimation of the SBP and DBP and should be a topic of future research. Systolic and diastolic BP values measured with a cuff-based BP monitor, as we outlined in the methods, could also be for personalization. Such methods to non-invasively measure direct BP might be inherently more noisy than intra-arterial blood pressure readings, and our models may need to be further refined accordingly.

Another important limitation of our approach is the need for further improvement in accuracy of the PPG-based approach based on the length of the test data segment as well as using data from a more diverse population. While we used a 10 s data segment for calibration, the optimal length of this data segment for calibration needs to be further investigated. One final limitation of our method involves filtering to clean the VitalDB data, which may have impacted the signal-to-noise ratio, contributing to the higher variation in DBP and SBP estimates. Because this data cleaning and filtering was already conducted from the shared VitalDB database, any methods that affected the signal-to-noise ratio would have propagated into our residual errors.

In conclusion, our research clearly showed that personalization using “ground truth” BP mean and SD values is a robust method for estimating future SBP and DBP reliably over a long period of time, while a calibration-free method without personalization is not. Personalization significantly reduces BP error variance in the short term as well as over a longer period to address long-term drift. This makes it clinically meaningful for commercial deployment over long time periods. We believe that the promising results from this current research will provide a strong impetus for future work in this area and will be the subject of our future research. In our future research, we will validate this method using the European Society of Hypertension protocol [9] as well as using alternate direct blood pressure methods for calibration, such as the cuff or TAG sensors. Our future research will also focus on personalized models for diseases such as cardiac morbidity, chronic renal failure, malignant and secondary hypertension, pre-eclampsia, and autonomic neuropathy, to name a few, that could have significant implications for quality of life and healthcare costs [3]. Such models based on high-quality real-world data from improved sensors, coupled with optimized ML or DL algorithms and personalization, may also elucidate other novel markers for improving diagnosis of diseases or improving clinical management of patients.

## Figures and Tables

**Figure 1 bioengineering-12-00493-f001:**
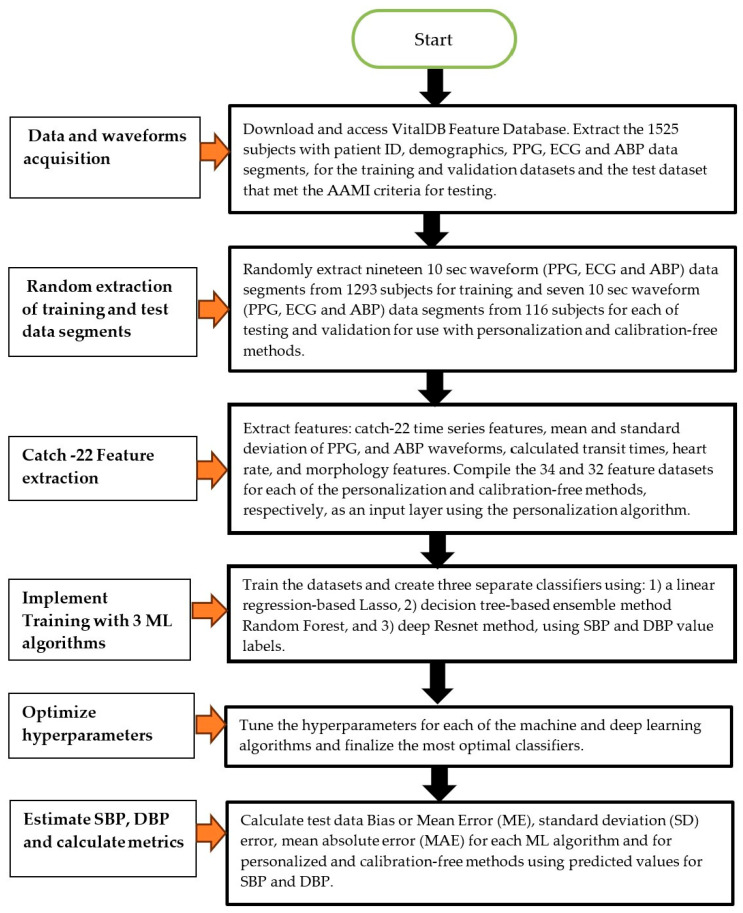
Schematic diagram of the utilized methodology.

**Figure 2 bioengineering-12-00493-f002:**
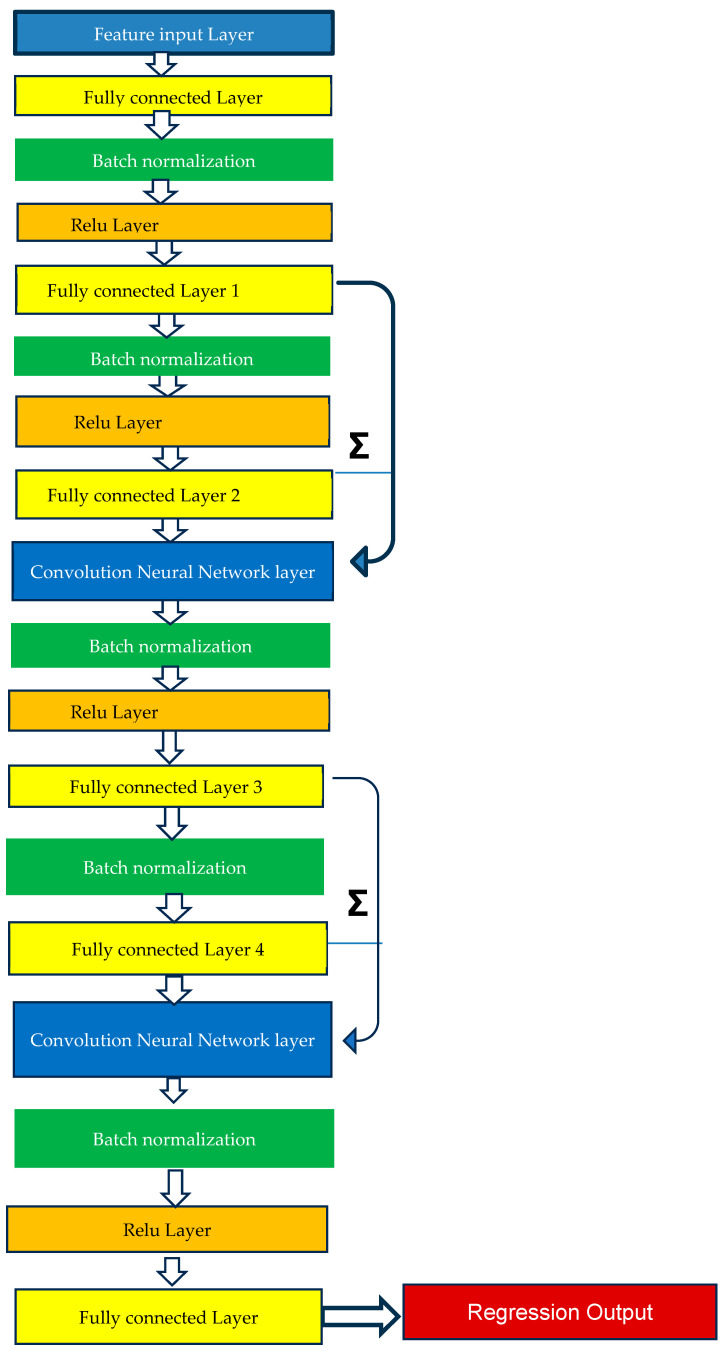
ResNET architecture.

**Figure 3 bioengineering-12-00493-f003:**
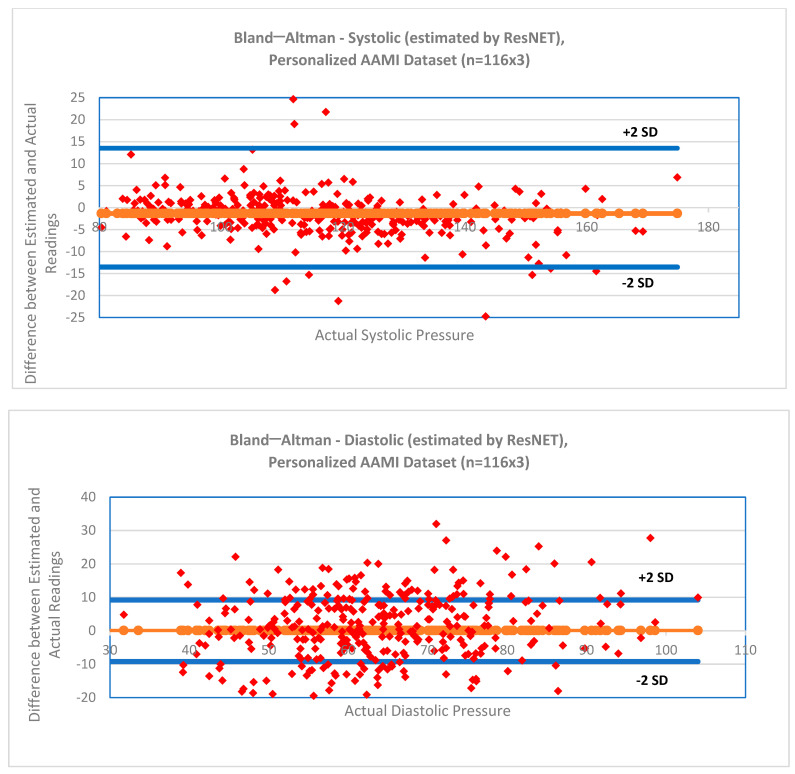
Per-segment bias and standard deviation estimation test set errors, personalized method using ResNET, n = 116 × 3. Blue lines show ± 2 standard deviations. a. SBP comparison. b. DBP comparison. Axis scales differ between plots.

**Table 1 bioengineering-12-00493-t001:** Literature review of various recent machine and deep learning methods used for BP estimation. MAE = mean absolute value.

Authors	Year	Sensor Signal	Method Used	Sample Size	Data Source	Study Duration	Mean ± SD Errors (mm Hg)
Tanveer et al.	[10]/2019	ECG, PPG	Artificial neural network-long short-term memory network	39 (hemodynamically compromised)	Prospective	Short term (40 s)	SBP: 1.10 (MAE) DBP: 0.58 (MAE)
Eom et al.	[11]/2020	PPG	Convolution neural network and bi-directional gated recurrent network	15 (healthy)	Prospective	Short term (<1 h)	SBP: −0.20 ± 5.83 DBP: −0.02 ± 4.91
Sadrawi et al.	[12]/2020	PPG	Genetic deep convolution autoencoder	18 (healthy, hemodynamically compromised)	Prospective	Short term (<6 h)	SBP: −1.66 ± 7.496 DBP: 0.66 ± 3.31
Athaya et al.	[13]/2021	PPG	U-net deep learning architecture	100 (hemodynamically compromised)	MIMIC, MIMIC III	Short term (3.4 h)	SBP: 3.68 ± 4.42 DBP: 1.97 ± 2.92 MAP: 2.17 ± 3.06
Jeong et al.	[14]/2021	ECG, PPG	Convolution neural network-long short-term memory combination	48 (healthy)	Prospective	Short Term (<800 s)	SBP: 0.2 ± 1.3 DBP: 0.0 ± 1.6
Fan et al.	[15]/2021	ECG	Bi-layer long short-term memory network	942 (hemodynamically compromised)	MIMIC II	Short Term (~230 s)	SBP: 7.69 ±10.83 DBP: 4.36 ± 5.90 MAP: 4.76 ± 6.47
Hu Q et al.	[16]/2022	PPG	Convolution neural network with attention mechanism, multi-task learning	1825 (hemodynamically compromised)	UC, Irvine database	Short Term (<20 min)	SBP: 0.97 ± 8.87 DBP: 0.55 ± 4.23
Ibtehaz et al.	[17]/2022	PPG	Two-stage cascaded convolution neural network	942 (healthy and hemodynamically compromised)	MIMIC III	Short Term (<30 min)	SBP: 5.7 ± 9.2 DBP: 3.4 ± 6.1 MAP: 2.3 ± 4.4
Jiang et al.	[18]/2022	ECG, PPG	Neural network with multi-task learning	3000 (hemodynamically compromised)	MIMIC-II	Short Term (60 h)	SBP: 4.04 ± 5.8 DBP: 2.29 ± 3.55 MAP: 2.46 ± 3.58
Mahmud et al.	[19]/2022	ECG, PPG	Shallow one-dimensional auto-encoder (U-net architecture)	942 (hemodynamically compromised)	MIMIC II	Short Term (<21 min)	SBP: 2.73 (MAE) DBP: 1.17 (MAE)
Seok et al.	[20]/2021	BCG	Convolution neural network	30 (healthy)	Prospective	Short Term (<10 s)	SBP: 0.93 ± 6.24 DBP: 0.21 ± 5.42
Treebupachatsakul et al.	[21]/2022	ECG, PPG	Fourier transformation followed by deep learning	>2500 (healthy and hemodynamically compromised)	Kachuee et al., 2015 [22]	Short Term (<30 min)	SBP: 7 DBP: 6
Mahardika et al.	[23]/2023	PPG, ABP	Convolution neural network, long short-term memory network	55	MIMIC-III	Short Term (<5 min)	SBP: 0.13 ± 7.04 DBP: 0.48 ± 3.79
Vliet et al.	[24]/2024	PPG	Machine learning algorithm—exact method not disclosed	124	Prospective	Short Term (<30 s)	SBP: ±3.7 ± 4.4 DBP: ±2.5 ± 3.7
Huang et al.	[25]/2024	BCG	Deep learning UUNet	40 (nighttime)	Kansas dataset	Short term (<30 min)	SBP: −0.19 ± 8.31 DBP: −0.04 ± 4.48
Liu et al.	[26]/2025	BCG, IPG	Random forest, XGBoost	17	Prospective (healthy)	Short term (<18 min)	SBP, MAD: 3.54 DBP, MAD: 2.57

**Table 2 bioengineering-12-00493-t002:** Comparison Table—Lasso, random forest, and ResNET.

	Lasso	Random Forest	ResNET
Type	Machine Learning	Machine Learning	Deep Learning
Principle	Based on least square multiple regression adjusted for overfitting	Based on an ensemble of decision trees based on bagging or boosting these trees	Based on multiple layers (18) of convolution neural networks with batch normalization and ReLU activation function
Cross-validation	Yes—10-fold	Yes—10-fold	Yes—5-fold
Complexity	Low	Medium	High
Hyperparameter	Alpha—Lasso to ridge ratio	Learning rate, leaf size, learning cycles, splits and features to sample	Number of epochs, learning rate
Computation time	Low	High	High
Input feature type	Binary, numerical	Binary, numerical	Binary, numerical, and categorical
Fit parameters	100	492	840
Data amount (>10 times fit parameter)	Can work with relatively less data	Needs more data for effective performance	Needs large amounts of data

**Table 3 bioengineering-12-00493-t003:** Subject demographics—VitalDB dataset (n = 1525).

Descriptor	Training Subjects	Validation Subjects	Test Subjects
Male	751 (58%)	69 (59%)	69 (59%)
Female	542 (42%)	47 (41%)	47 (41%)
Age (>40 years)	1137 (88%)	103 (89%)	103 (89%)
Total	1293	116	116

**Table 4 bioengineering-12-00493-t004:** (**a**) Calibration-free method errors using the AAMI standard—bias (μ) ± standard deviation (SD) and mean absolute error (MAE) from each machine learning model derived from the estimated SPB and DBP using the calibration-free method for the test dataset (n = 348 or 116 × 3) extracted from VitalDB. All errors are in mm Hg. (**b**) Personalized method errors using AAMI standard—bias (μ) ± standard deviation (SD) and mean absolute error (MAE) from each machine learning model derived from estimated SPB and DBP using the personalized method for the test dataset (n = 348 or 116 × 3) extracted from VitalDB. All errors in mm Hg. (**c**) Performance comparison with the British Hypertension Society (BHS) standard (n = 348).

(**a**)						
	**Lasso**		**Random Forest**		**ResNET**	
	**μ ± SD**	**(MAE)**	**μ ± SD**	**(MAE)**	**μ ± SD**	**(MAE)**
SBP	−2.11 ± 18.78	14.25	−1.08 ± 18.76	14.20	−1.86 ± 19.55	14.69
DBP	−0.68 ± 11.77	9.18	−0.24 ± 11.06	8.56	0.11 ± 11.55	9.06
(**b**)						
	**Lasso**		**Random Forest**		**ResNET**	
	**μ ± SD**	**(MAE)**	**μ ± SD**	**(MAE)**	**μ ± SD**	**(MAE)**
SBP *	−1.51 ± 8.04	4.88	−1.32 ± 7.97	4.95	−1.31 ± 7.91	4.83
DBP *	−0.52 ± 4.69	2.62	−0.43 ± 4.62	2.60	0.10 ± 4.59	2.65
(**c**)						
			**Absolute Difference**			
			**≤5 mm Hg**	**≤10 mm Hg**	**≤15 mm Hg**	**Grade**
BHS Standard		SBP, DBP	60%	85%	95%	A
			50%	75%	90%	B
			40%	65%	80%	C
			Worse than C			D
Proposed Model: Personalized, ResNET		SBP	65.8%	91.7%	95%	A
		DBP	85.9%	96.8%	98.6%	A

* All values for mean error and SD are within the AAMI standard limits for bias or an ME of 5 mm Hg and for an SD of 8 mm Hg with the random forest and ResNET methods. Note: The personalization method used BP information from the test subject data and transferred this learning to the general trained algorithm layers.

**Table 5 bioengineering-12-00493-t005:** Levene’s test: F-test and *p*-value results for significant differences in variances for SBP and DBP estimation errors between 3 ML algorithms and 2 calibration methods (n = 116).

Method	SBP	DBP
	F-Statistic, *p*-Value	F-Statistic, *p*-Value
Machine Learning Algorithm	F(1) = 0.00, 0.99	F(1) = 0.19, 0.82
Method (Calibration-free vs. Personalized)	F(2) = 45.9, **0.00**	F(2) = 46.9, **0.00**

Significant difference (in bold).

**Table 6 bioengineering-12-00493-t006:** Summary results of “ground truth” evaluation and statistical comparisons between ground truth and values estimated using the ResNET algorithm (n = 116).

Method	Systolic BP Estimation Errors			Diastolic BP Estimation Errors		
	Bias	SD	MAE	Bias	SD	MAE
Estimated by algorithm (in mm Hg)	−1.31	7.91	4.83	0.10	4.59	2.65
Estimated using ”ground truth” (in mm Hg)	−1.21	19.46	14.75	−0.40	13.68	9.87
*p*-values and F-statistic, bias (ANOVA)	F = 1.28, *p* = 0.26			F = 2.06, *p* = 0.15		
*p*-values and F-statistic, variance (Levene’s test)	F = 91.42, ***p* = 0.00**			F = 99.04, ***p* = 0.00**		

Significant difference (in bold). SD—Standard deviation. MAE—Mean absolute error.

## Data Availability

The data are available upon request from Rajesh S Kasbekar, rkasbekar@gmail.com, Department of Biomedical Engineering, Worcester Polytechnic Institute, Worcester, MA, USA.

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
