# Peer review of "Optimizing Input Feature Sets Using Catch-22 and Personalization for an Accurate and Reliable Estimation of Continuous, Cuffless Blood Pressure"

_bioengineering, 2025, doi:10.3390/bioengineering12050493_

Round 1
Reviewer 1 Report
Comments and Suggestions for Authors
In the manuscript titled “Optimizing input feature sets using catch-22 and personalization for accurate and reliable estimation of continuous, cuffless blood pressure”, the authors focused on optimizing machine learning algorithms using the catch-22 features, when applied to photoplethysmogram waveform and personalized with direct BP data. The focus of the research work was to study and explore the impact of scale as well as feature selection and personalization in enhancing the BP estimates. The paper my be considered for publication after responding to the following queries.
Major:
- L.145: How and why the data cleaning was required when Wang et al [27] have already pre-processed the data? Which samples are invalid and why?
- Although reference of catch-22 feature extraction algorithm is given, however, a brief description/explanation of this should be given in the text for understanding of the reader.
- The data used and the code should be provided that can reproduce the given results.
- L 165-166: Why and how Per subject 19, 7 and 7 “10-s segments” for each of training, validation and testing, were selected?
- How the hyperparameters for machine and deep learning algorithms were tuned for optimization?
- Why the elastic net hyperparameter “alpha” is fixed at 0.75? Its optimal value can easily be calculated and should be decided on the basis of cross-validation.
- Was lasso used independently on training, validation and test data for feature selection? If yes, the selected features can differ. If no, then it reflects that validation and test data are not being handled independently. How all these issues were handled and addressed?
- While using Random Forest, again, how feature selection was performed on training, validation and test data sets. How these selected features were different from lasso?
- For ResNet, using convolutional neural network with18 layers will make the system extremely complex and is it reasonable to use such a complex model with 32 or 34 features only. Also, how it was implemented on training, validation and test data.
- In all three used methods, important features should be highlighted like in lasso using lasso path, Gini importance in random forests etc.
- The results of randomly splitting the data into training, validation and test data only once can differ if the same random splitting is replicated. One can replicate such splitting and can take the average values to avoid any misleading results.
- What is the rationale behind “conditional testing” in Section 2.5?
- The measures ME, MAE, and SD error, residual error should be defined in the text before their use.
- Why Levene’s test was used and importance of its results need to be discussed with reference to the current work. The discussion on L322-26 needs clarification for the reader about its effectiveness and what is actually its results are conveying.
Minor:
The paper should be carefully reconsidered for typos and grammatical errors. Some examples are:
- Some correction needed at L. 145 …. [sections 2.4 and 2.5, 27] ….
- L194: … Figure 1 shows a schematic of …. Should be “…Figure 1 shows a schematic diagram of…”
The paper should be carefully reconsidered for typos and grammatical errors. Some examples are:
- Some correction needed at L. 145 …. [sections 2.4 and 2.5, 27] ….
- L194: … Figure 1 shows a schematic of …. Should be “…Figure 1 shows a schematic diagram of…”
Author Response
Major:
- L.145: How and why the data cleaning was required when Wang et al [27] have already pre-processed the data? Which samples are invalid and why?
Agree. The data extraction, filtering and cleaning mentioned in L.145 was done by Wang et al. This is now indicated in L.165 and L.166 of the revised document (clean copy) as follows:
The criteria used by Wang et al. [30] for extraction included presence of ECG lead-II, fingertip PPG and ABP signals. Wang et al. extracted, filtered and cleaned data for invalid samples, saturated and flatline signals, and signal quality (see [30], sections 2.4 and 2.5).
However, note that these signals sometimes had less than 3 peaks for the PPG and ECG signals. To ensure that we have data records with signals having 3 peaks or more as indicated in L.214 through 220, we had to retain only those waveforms that had at least 3 peaks for these signals. So, any data segments with less than 3 peaks are invalid, since we used an average of 3 peaks to derive our temporal and morphology features. We added this clarifying line on L.179.
We, further removed the ECG, PPG and ABP signals with less than 3 peaks ….
2. Although reference of catch-22 feature extraction algorithm is given, however, a brief description/explanation of this should be given in the text for understanding of the reader.
Response: Thanks for bringing this to our attention. We have added this text on page 3-4 in L-107 through111 in the introduction to clearly explain what catch-22 is.
Catch-22 is a relatively small set of relevant features from the pantheon of features utilized in time-series analysis. This small set of features (including linear and non-linear autocorrelation, successive differences, value distributions and outliers, and fluctuation scaling properties) has exhibited strong classification or regression performance in past studies and is minimally redundant.
3. The data used and the code should be provided that can reproduce the given results.
The data are already publicly available from the original authors, as cited in the manuscript. The software code is now included in the supplementary material. Note that due to the proprietary nature of some code, we have provided the logic only.
4. L 165-166: Why and how Per subject 19, 7 and 7 “10-s segments” for each of training, validation and testing, were selected?
Response:
Thanks for bringing this to our attention. We added the following to lines L190-L199 of the revised document.
“The 19 segments were randomly chosen since for 1293 subjects they gave a total of 24,567 data segments which were well above fit parameter requirements (10 times 840 fit parameters or 8400) of the deep learning ResNET algorithm. As detailed below, only 3 test values per subject were evaluated for estimation error from the 7 available test segments as stipulated by the AAMI criteria. Seven segments were randomly chosen and arranged in numeric order because the first segment was used for personalization, 3 test values per subject were required by the AAMI criteria and 3 additional data segments had to be extracted to maintain independence of the direct pressure calibration values if the flow criteria were not met in the initial 3 segments as explained in section 2.5 below. “
Training: We had a data set of 1293 subjects with hundreds of 10-second data segments. We randomly extracted 19 ‘10-second data segments’, because it gave us a total of 1293 x 19 or 24,567 data elements. Since we used the ResNET deep learning algorithm, one of our goals from our previous publication was to apply the deep learning algorithm on a large dataset. We estimated that a total of 24, 567 data segments used with 32 features would be adequate data (10 times the 840 fit parameters for the ResNET deep learning algorithm) to obtain adequate resolution to do the training using the ResNET deep learning algorithm.
Test and Validation: We had a data set of 116 subjects. The 1st data segment was used for personalization. The AAMI standard requires you to test 3 separate data segments for every subject (for a total of at least 85 x 3). However, since our personalization algorithm used a PPG flow-criteria to do the personalization, we had to take an additional 3 data segments to maintain independence if the flow criteria were not met in the initial 3 segments. Hence, we have a total of (1+3+3) or 7 data segments.
5. How the hyperparameters for machine and deep learning algorithms were tuned for optimization?
Response
Thanks for bringing this to our attention.
Lasso and ResNET:
The hyperparameters for the Lasso and ResNET deep learning algorithms were based on several iterations using the validation dataset. For Lasso, we varied the alpha hyperparameter in increments of 0.05 from 0.5 to 1 in a previous study, and have referred readers to this prior study within the manuscript. A value of 0.75 gave us the most optimal results. Similarly, the hyperparameters for the ResNET algorithm were derived using various iterations of the validation dataset for learning rate, number of epochs, minimum batch size and validation frequency. The optimal values are listed in the manuscript and were found to be 0.001 for learning rate, and 50 for epochs, batch size and validation frequency.
Random forest
The built-in function in MATLAB ['OptimizeHyperparameters’, 'all'] was used for optimizing the hyperparameters for the Random Forest. This function cycled through all the lag and boost iterations as well as the learning cycles, learning rate, minimum leaf size and maximum number of splits to identify the most optimal combination of hyperparameters for the training dataset.
This is outlined in section 2.4 L-269-276 of the revised manuscript.
6. Why the elastic net hyperparameter “alpha” is fixed at 0.75? Its optimal value can easily be calculated and should be decided on the basis of cross-validation.
Response:
We did a 10-fold cross validation while training using the Lasso algorithm. It could be decided based on cross-validation; however, we chose to fix it after doing several iterations on the validation dataset in a prior related study [23] where we varied the value of alpha in increments of 0.05 from 0.5 to 1. These iterations yielded an optimal value of 0.75.
We added the following to the revised document in L255 to L.256.
For our Lasso models (Lasso command in MATLAB), BP prediction was based on a linear model using least squares regression and 10-fold cross validation. The elastic net hyperparameter ("alpha") was selected as 0.75 based on our prior experience [26].
7. Was lasso used independently on training, validation and test data for feature selection? If yes, the selected features can differ. If no, then it reflects that validation and test data are not being handled independently. How all these issues were handled and addressed?
Response:
The selection of features was based on our extensive past experience for clinical relevance of the features, literature review and an exploration of the most relevant features. The catch-22 algorithm also extracted 22 typically relevant features. Lasso was used independently on the training dataset using these features to derive Lasso coefficients and not for feature selection. It was then applied on the independent test and validation datasets to predict the blood pressure values using the same features used in training as the input features. This also simulates real world scenarios where the test data are not always available for feature selection.
For clarity, note that we added on L.239 and 240 of revised document to the Methods section: “All testing used data that was independent from training and validation data.” This statement is provided prior to the details of all models and, thus, applies to all models.
8. While using Random Forest, again, how feature selection was performed on training, validation and test data sets. How these selected features were different from lasso?
Response: We used the same set of features (32 for calibration free and 34 for personalized method) in all the three algorithms. This was important because we could then compare the performance of each algorithm with each other based on the same input features using the ANOVA and Levene’s test.
The feature set used in the Random Forest algorithm consisted of demographic, temporal, morphology and catch 22 features for a total of 32 features in the calibration free method. In the personalization method, we used two additional features as explained in section 2.3 on feature extraction, L 207 to L.228 of the revised manuscript. The features were chosen based on extensive experiments and use of shapely analysis (a method to identify the most relevant features) to identify the features as explained in #10.
We added the following to line 225 and 226 of the revised manuscript:
These same sets of features were derived from clinical relevance as well as catch-22 feature extraction and then used in all three ML algorithms, to enable a comparison of the performance among the three algorithms.
9. For ResNet, using convolutional neural network with18 layers will make the system extremely complex and is it reasonable to use such a complex model with 32 or 34 features only. Also, how it was implemented on training, validation and test data.
Response:
The ResNET algorithm works well when you have a large amount of data. In our case a total of 19 separate data records from1293 subjects gave a total of 24,567 line item data segments in the training matrix. Using 32 or 34 features on these 24,567 data segments gave 786,144 data elements for doing the training. This is a large amount of data and therefore the ResNET algorithm worked quite well on this data. It was implemented using the stochastic gradient descent algorithm from the MATLAB deep learning toolbox, where the 18 layers consisting of CNN’s, batch normalization layers, ReLU layers and Input and Output layers were used to train and create a ResNET classifier using the training data. The validation data was used to optimize the hyperparameters using the mean square error and the loss function. After using several iterations, we found the learning rate of 0.001, a batchsize of 256 and an epoch size of 50. The resulting classifier was then used to predict the systolic and diastolic pressure using the independent test dataset that met the requirements of the AAMI criteria.
10.In all three used methods, important features should be highlighted like in lasso using lasso path, Gini importance in random forests etc.
Response:
Thanks for bringing this to our attention. We added the following on line 107 to 110 of the revised document:
Catch-22 is a method to extract a relatively small set of relevant features from the pantheon of features utilized in time-series analysis. This small set of features (includes linear and non-linear autocorrelation, successive differences, value distributions and outliers, and fluctuation scaling properties) has exhibited strong classification or regression performance in past studies and is minimally redundant.
We also added the following on lines 225 to 226 of the revised document:
These same sets of features were derived from clinical relevance as well as catch-22 feature extraction and then used in all three ML algorithms, to enable a comparison of the performance among the three algorithms.
We extracted the 34 features out of the original 69 features based on the most relevant feature extraction using catch 22 and a shapely analysis of the classifier. The catch 22 features extracted 22 features that were based on the relevance of these features. The remaining features (demographic, morphology and temporal) were based on clinical relevance and our previous experience. Hence all the 34 are important. Lasso path and Gini are alternate methods, but the feature extraction we used was comprehensive.
11. The results of randomly splitting the data into training, validation and test data only once can differ if the same random splitting is replicated. One can replicate such splitting and can take the average values to avoid any misleading results.
Response: This is a great question. We did not do the training on data that was split just once. In fact, the replication of the splitting was achieved using a 10-fold cross-validation using the training data for the lasso and random forest algorithm, and by using a 5- fold cross validation using the resnet algorithm. We have updated the manuscript to reflect this in L256, L-268 and L-287 on page 7 and 8 of the revised manuscript. The model built on this cross-validation data was then used to predict using the independent test data set as well as the validation dataset. The AAMI standard requires you to do the testing on only 1 set of test data.
We added the following to lines 190-L.192 to explain the fit parameter requirements for ResNET:
‘The 19 segments were randomly chosen since for 1293 subjects they gave a total of 24,567 data segments which were well above fit parameter requirements (10 times 840 fit parameters or 8400) of the deep learning ResNET algorithm.’
12. What is the rationale behind “conditional testing” in Section 2.5?
Response: Thanks for bringing this to our attention. The rationale behind the ‘conditional testing’ is to use the PPG flow or the Pulse Arrival time signal as the trigger for calibration or personalization. This is the key innovation of our algorithm. The estimation of the blood pressure is done using the machine learning algorithm as long as the flow is within 5% of the previous values. If the flow changes by over 5%, it automatically triggers a personalized calibration. This ensures that the variance of the readings are reduced to a level where they can meet the 8 mm Hg requirement of the AAMI standard. Without this improvisation, the confounding factors affecting the pressure-flow relationship play a major role in increasing the variance of the readings as seen in the standard deviation of the calibration-free method which exceeds the 8 mm Hg limit.
Please see the attached text in L.123 to L141 of the revised document:
The European Society of Hypertension working group on blood pressure monitoring and cardiovascular variability issued a statement in 2022 [7] recommending at that time against adoption of devices estimating BP using cuffless methods in clinical practice……
Our key contribution, therefore, is based on the premise that the SBP and DBP, estimated non-invasively by our method, is reliable only if the flow or the pulse arrival time is within a certain quantum band. A change in flow or pulse arrival time outside this band triggers a personalization or calibration that would ensure that accuracy is preserved.
13. The measures ME, MAE, and SD error, residual error should be defined in the text before their use.
Response: Thanks for bringing this to our attention. We have defined these and added them in L.326 to L.329 in the manuscript.
14. Why Levene’s test was used and importance of its results need to be discussed with reference to the current work. The discussion on L322-26 needs clarification for the reader about its effectiveness and what is actually its results are conveying.
Response: We appreciate this comment. We added the following text L-384 to 393 on page 12-13 to the revised manuscript:
Levene’s test is a statistical method to assess the difference between the variances of two independent datasets. It uses the mean absolute error to test the null hypothesis that the variances of two datasets estimated using different methods or using different algorithms (also referred to as homogeneity of variance or homoscedasticity) are equal. A p-value less than 0.05 indicates a difference between the variances in the datasets assuming a normal distribution and random sampling. Our objective in using this test was i) to infer statistically whether the variance of the data estimated using the personalization method differed from the variance of data estimated using the calibration-free method, and ii.) to infer statistically whether the variance of the data estimated using the three ML algorithms varied from each other.
Minor:
The paper should be carefully reconsidered for typos and grammatical errors. Some examples are:
- Some correction needed at L. 145 …. [sections 2.4 and 2.5, 27] ….
Response: Please see the corrected text on L.165-168 of revised document as follows:
The criteria used by Wang et al. [30] for extraction included presence of ECG lead-II, fingertip PPG and ABP signals. Wang et al. extracted, filtered and cleaned data for invalid samples, saturated and flatline signals, and signal quality (see [30], sections 2.4 and 2.5).
2. L194: … Figure 1 shows a schematic of …. Should be “…Figure 1 shows a schematic diagram of…”
Response: Added diagram after schematic on L.228 on page 6 of revised document:
Figure 1 shows a schematic diagram of the flow steps consisting of feature extraction, training, hyperparameter selection and testing.
Comments on the Quality of English Language
The paper should be carefully reconsidered for typos and grammatical errors. Some examples are:
- Some correction needed at L. 145 …. [sections 2.4 and 2.5, 27] ….
Response: See minor comment 1 above.
2. L194: … Figure 1 shows a schematic of …. Should be “…Figure 1 shows a schematic diagram of…”
Response: See minor comment 2 above.
Response: We also went through the whole manuscript again and made changes on the following lines in the revised document for typos and grammar. We also used MDPI editing service.
L.153 : changed ‘de-identified’ to ‘deidentified’
L.170: changed ‘minima’ to ‘minimum’
- 211: changed ‘Heart Rate’ to ‘The heart rate…’
L.237: changed ‘prediction’ to ‘predictions’
Table 4.b (L.369, page 11): changed ‘the BP’ to ‘BP’
Line.377: changed ‘Two-way’ to ‘A two-way…’
Line 379: changed ‘paired multiple comparisons’ to ‘paired multiple-comparisons’

Reviewer 2 Report
Comments and Suggestions for Authors
The work is well organized and written. The contribution of the study to the literature should be clearly written in one paragraph only in the introduction. I believe that it is appropriate to publish the study after this change.
Author Response
1. The work is well organized and written. The contribution of the study to the literature should be clearly written in one paragraph only in the introduction. I believe that it is appropriate to publish the study after this change.
Response: Thanks for bringing this to our attention. We have added the contribution of this study to literature in the introduction section of the revised manuscript (L.123 to L.141) as follows:
The European Society of Hypertension working group on blood pressure monitoring and cardiovascular variability issued a statement in 2022 [7] recommending at that time against adoption of devices estimating BP using cuffless methods in clinical practice. According to their consensus statement, although cuffless sphygmomanometers are a promising technology, standardization of accuracy, tracking of dynamic variations and suitability for long-term use is required—and the existing literature had not demonstrated these characteristics. For widespread use in clinical practice, evidence is needed not only for measurement in healthy subjects, but also in pre-hypertensive and hypertensive patients, to ensure that the system can be used in the same way as, or instead of, current blood pressure monitors. Therefore, until a device is developed that can meet these requirements, its use in clinical practice cannot be recommended. We adopted a different approach to address this exact issue. Our key contribution, therefore, is based on the premise that the SBP and DBP, estimated non-invasively by our method, is reliable only if the flow or the pulse arrival time is within a certain quantum band. A change in flow or pulse arrival time outside this band triggers a personalization or calibration that would ensure that accuracy is preserved. Any creep due to confounding factors contributing to a large BP variance, especially over longer time periods, would no longer be relevant in increasing such inherent variance. This ensures that the accuracy and reliability of the BP estimates is maintained over the long term for hypertensive, healthy or hypotensive subjects and therefore could be suitable for clinical adoption.

Reviewer 3 Report
Comments and Suggestions for Authors
The study highlights the importance of nighttime monitoring of cuffless blood pressure (BP) for the prognosis of cardiovascular and other diseases due to its strong predictive ability. We studied how optimised machine learning using capture-22 features can predict systolic and diastolic BP when applied to the photoplethysmogram waveform and personalised directly with BP data via transfer learning. The study contributes to the literature in terms of subject matter and proposed methods. However, some minor modifications are required.
1. There are many machine learning and deep learning approaches (Lasso, Random Forest, and Resnet). What is the rationale for choosing these architectures? Give a comparison table.
2. Table 1, which gives a summary of the studies in the literature, covers studies between 2019-2023. If there are studies conducted in 2024 and 2025, provide a more up-to-date table by adding them.
3. For a more streamlined table, it is recommended to use (Table 1) author/year. (Tanveer et al.[10]/2019)
4. When the methods are given, Lasso and Random Forest are explained very briefly and only a drawing of the architecture of the RESNET model is given. Please elaborate on the Lasso and Random Forest algorithms.
5. The year information in the citations is bold in some citations and not in others. For the sake of completeness, give them in the same format
6. ‘Appendix 1- Catch-22 detailed description. ‘ table is given as an appendix. Please refer to the text in which the Catch-22 detailed description table is given in Appendix 1.
Author Response
- There are many machine learning and deep learning approaches (Lasso, Random Forest, and Resnet). What is the rationale for choosing these architectures? Give a comparison table.
Response: Agree. We added the following table 2 (L.245) to the manuscript along with rationale for choosing these architectures. We also added the following text explaining the rationale for selection L.235-L.237, page 7:
These model forms include varying levels of complexity and number of fit parameters (Lasso having the lowest and ResNET the highest).
Table 2. Comparison Table- Lasso, Random Forest and ResNET
|
Lasso |
Random Forest |
Resnet |
Type |
Machine Learning |
Machine Learning |
Deep Learning |
Principle |
Based on least square multiple regression adjusted for overfitting |
Based on an ensemble of decision trees based on bagging or boosting these trees |
Based on multiple layers (18) of Convolution neural network with batch normalization and ReLU activation function |
Cross validation |
Yes – 10-fold |
Yes- 10-fold |
Yes- 5-fold |
Hyperparameter |
Alpha – lasso to ridge ratio |
Learning rate, leaf size, learning cycles, splits and features to sample |
Number of epochs, learning rate |
Computation Time |
Low |
High |
High |
Input Feature Type |
Binary, numerical |
Binary, numerical |
Binary, numerical and categorical |
Fit parameters |
100 |
492 |
840 |
Data amount (>10 times fit parameter) |
Can work with relatively less data |
Needs more data for effective performance |
Needs large amount of data |
Table 1, which gives a summary of the studies in the literature, covers studies between 2019-2023. If there are studies conducted in 2024 and 2025, provide a more up-to-date table by adding them.
Response: Agree. We have added studies in 2024 and 2025 in Table 1, L.92, page 3 and referenced them in the references section on page 16 (L.578-L.584).
- For a more streamlined table, it is recommended to use (Table 1) author/year. (Tanveer et al.[10]/2019)
Response: Agree. We have updated the table to include author/year as suggested. Please see revised manuscript L.92 onwards.
When the methods are given, Lasso and Random Forest are explained very briefly and only a drawing of the architecture of the RESNET model is given. Please elaborate on the Lasso and Random Forest algorithms.
Response:
Agree. We have elaborated on the Lasso and RF algorithms along with a rationale for their selection and a comparison table. Please see page 7 and 8 L.246 to L.254 and L.260 to L.266 for detailed description.
Lasso (least absolute shrinkage and selection operator) is a regression analysis method that chooses key features and performs regularization to prevent overfitting and to enhance prediction accuracy and interpretation of the resulting model. Lasso extends the ordinary least squares regression by adding a penalty term to the regression equation. The penalty term is the sum of the absolute values of the coefficients, which helps in shrinking some coefficients to zero, effectively performing feature selection. Lasso uses an elastic net hyperparameter ("alpha"), an estimate of the Lasso to ridge variance, which controls the strength of the penalty.
…
Random forest is an ensemble learning method (which we used for regression) that works by bagging and averaging a multitude of decision trees using a random subset of features from the main feature set. The training algorithm applies the bagging technique to tree learners as follows: Given a training set with multiple responses, bagging repeatedly “n” times selects a random sample with replacement of the training set and fitting trees to these samples. After these “n” trees are trained, the predictions are averaged from all the individual regression trees.
The year information in the citations is bold in some citations and not in others. For the sake of completeness, give them in the same format
Response: Agree. We have updated the citations to convert the bold years in some citations to non-bold format. Please see L526 onwards in revised manuscript.
- ‘Appendix 1- Catch-22 detailed description. ‘ table is given as an appendix. Please refer to the text in which the Catch-22 detailed description table is given in Appendix 1.
Response: Agree. The summary text is reproduced at the bottom in the legend for Appendix 1. However, we added reference 30 and a statement pointing to section 2.4 within this reference: See section 2.4, paragraph 1 for details. See L.628 on page 18 of revised document. Also, note that Appendix 1 is reference in the main manuscript text in the first sentence L.201 of section 2.3 on page 5.
